# Interactions between Macrophages and Mast Cells in the Female Reproductive System

**DOI:** 10.3390/ijms23105414

**Published:** 2022-05-12

**Authors:** Nadia Lampiasi

**Affiliations:** Consiglio Nazionale delle Ricerche, Istituto per la Ricerca e l’Innovazione Biomedica, Via Ugo La Malfa 153, 90146 Palermo, Italy; nadia.lampiasi@irib.cnr.it; Tel.: +39-0916809513; Fax: +39-0916895548

**Keywords:** innate immunity, mast cells, macrophages, polarization, cytokines, inflammation, peritoneum, uterus, menstruation, pregnancy

## Abstract

Mast cells (MCs) and macrophages (Mϕs) are innate immune cells that differentiate from early common myeloid precursors and reside in all body tissues. MCs have a unique capacity to neutralize/degrade toxic proteins, and they are hypothesized as being able to adopt two alternative polarization profiles, similar to Mϕs, with distinct or even opposite roles. Mϕs are very plastic phagocytic cells that are devoted to the elimination of senescent/anomalous endogenous entities (to maintain tissue homeostasis), and to the recognition and elimination of exogenous threats. They can adopt several functional phenotypes in response to microenvironmental cues, whose extreme profiles are the inflammatory/killing phenotype (M1) and the anti-inflammatory/healing phenotype (M2). The concomitant and abundant presence of these two cell types and the partial overlap of their defensive and homeostatic functions leads to the hypothesis that their crosstalk is necessary for the optimal coordination of their functions, both under physiological and pathological conditions. This review will examine the relationship between MCs and Mϕs in some situations of homeostatic regulation (menstrual cycle, embryo implantation), and in some inflammatory conditions in the same organs (endometriosis, preeclampsia), in order to appreciate the importance of their cross-regulation.

## 1. Development and Tissue Distribution

In order to adapt to environmental challenges, multicellular organisms have developed sentinel cells that are able to sense and react to environmental signals. Among these cells, present in every tissue of the organism, are Mϕs and MCs, two distinct cell populations of myeloid origin that, in vertebrates, are co-present in all body tissues and in barrier tissues (such as mucosal surfaces). They have a wide range of partially overlapping functions encompassing tissue homeostasis/repair/remodeling and defense (including the detoxification of venoms and toxins in the case of MCs) [1]. Mϕs are plastic and versatile cells, which can react to signals in the microenvironment by adopting a specific functional phenotype (polarization). More recently, in tumors, functional polarization was also envisaged for MCs, which can depend on the inflammatory microenvironment sustained by the functional presence of tumor-associated macrophages (TAMs) [2]. Indeed, in some tumors, MCs can have an anti-inflammatory phenotype and a protective role, whereas in others, they can have a pro-inflammatory phenotype and a tumorigenic role [3,4].

Mammalian adult immune cells derive from hematopoietic stem cells (HSCs) in the bone marrow (BM). However, long-lived tissue-resident innate cells (including Mϕs, MCs, and innate lymphoid cells) may be composed of cells from different ontogenies (fetal, adult), depending on the tissue. Fetal hematopoiesis occurs in three sequential waves that differ in time and space but partially overlap: the primitive, the transient (definitive), and the definitive wave. The primitive wave generates erythrocytes, megakaryocytes [5], and the first Mϕs that colonize the embryo and form brain-resident microglia [6]. MCs, Mϕs, monocytes, and granulocytes, seeded during the transient (definitive) wave, derive from fetal erythro-myeloid progenitors (EMPs) that are resident in the yolk sac (YS) [7,8,9] (Figure 1). During prenatal life, EMPs migrate in the fetal liver and in BM, where they expand, generating multipotent progenitors (MPs), and finally differentiate into circulating pre-macrophages (pMacs). Then, these cells migrate in tissues, undergo differentiation (F4/80^+^), and can self-renew independently from the pool of definitive HSCs [10]. On the other hand, in prenatal life, MPs that are resident in fetal liver, BM, and skin generate MC-committed-progenitors [8]. These cells can develop into MCs in local organs under the influence of local cytokines, and of stem cell factors (SCFs), independently from the definitive lifelong pool of HSCs, but they become progressively replaced by definitive MCs at later stages of development (except in the skin).

Finally, the definitive wave of hematopoiesis gives rise to immature-adult HSCs in the aorta-gonads-mesonephros (AGM) region in mice and humans [10] (Figure 1). These immature HSCs transiently colonize the fetal liver, the main hematopoietic organ during embryonic development, and ultimately in the BM, generating mature-adult HSCs and progenitor cells that, in the postnatal life, can differentiate into monocytes or Mϕs [11] and MCs [12,13]. 

Therefore, in many adult tissues, Mϕs and MCs are a mixture of cells with double hematopoietic origin, embryonic cells that are derived from YS and seeded during fetal life, and BM-derived adult cells that exit the bloodstream to repopulate the resident cell pool under pathological as well as inflammatory homeostatic conditions [8,14]. Embryonic tissue-resident and adulthood Mɸs from tissue-infiltrating monocytes play distinct roles, both in health and disease (see physiological and inflammatory role) [15,16]. 

## 2. General Biological Functions

MCs and Mϕs represent the first line of host defense, and for this reason, they are often positioned at the interphase between the tissue and the environment. Thus, they mainly reside near blood vessels or nerves and in the mucosal epithelia in the lung, gastrointestinal, and genitourinary tracts [17,18,19]. In some species, including rodents and humans, MCs and Mϕs are also placed within mesothelium-lined cavities, such as the peritoneal and the pulmonary cavities [20]. 

As major secretory cells, upon activation, MCs produce and release numerous biologically active substances, some of which are pre-formed and stored in the granules for rapid release (histamine, TNF-α, heparin, lysosomal hydrolases, and proteases), whereas others are synthesized de novo (TNF-α, IL-6, IL-13, IL-1, IL-5, GM-CSF, leukotrienes, and prostaglandins) [13] (Figure 2). Interestingly, human MCs can form new granules after degranulation, and thus participate in further subsequent degranulation events [21]. In addition, depending on the stimuli, MCs can secrete the contents of single granules (histamine), some granular contents without degranulation (histamine, heparin), or can selectively release a combination of specific mediators (serotonin, eicosanoids, TNF-α, and IL-6) [22]. MCs display steady-state granule release under normal physiological conditions, aiming at maintaining homeostasis by providing small levels of bioactive mediators. Conversely, activated MCs rapidly secrete de novo synthesized and pre-formed granules [21]. 

MCs participate in allergic reactions as they express high-affinity receptors for IgE (FcεRI) that bind the antigen (allergen) [13] (Figure 2); they can support neuroinflammation and neurogenic pain through the binding of neuropeptide Substance P (SP) to the cognate receptor Mas-related G protein-coupled receptor (MRGPR) [23], (Figure 2 and Table 1) resulting in degranulation within minutes and the release of histamine, IL-6, and TNF-α [24]. MCs are activated by LPS through TLR4, which triggers the production of IL-6 and TNF-α [25], and also through TLR2 [26], which can suppress MC activation and degranulation [26,27], or can promote IgE-mediated MC activation, even in the absence of degranulation [28,29,30] (Figure 2 and Table 1). 

In addition to those listed, MCs participate in multiple physiological as well as pathological responses, such as antigen-presenting cells, the control of vascular tone and permeability (histamine, heparin, and edema), neovascularization (histamine, serotonin, TNF, kinins, proteases, LTs, PGs, vascular endothelial growth factor (VEGF), and platelet activated factor (PAF)), innate bacterial clearance (recruitment of neutrophils to the site of infection), neutralization of venoms (detoxification with proteases), and can contribute to the resistance of parasite infections. Furthermore, MCs can control the intensity and duration of the immune response by regulating the recruitment, survival, differentiation, phenotype, and ultimately the function of the immune cells involved in the response, through the numerous cytokines produced (i.e., IL-10, IL-6, IL-1, and TNF-α) [20]. 

Based on their protease content, MCs can be distinguished in at least two major populations in humans and mice, which differ in both tissue location and function [17]. One type contains tryptase only, (named mucosal mast cells (MMCs) in mice and MC_T_ in humans), while the other type contains tryptases, chymases, carboxypeptidase A, and cathepsin G (named connective tissue mast cells (CTMCs) in mice and MC_TC_ in humans). CTMCs are considered to be innate or constitutive MCs, and they are identified as being unique source of heparin, whereas MMCs are smaller, often hypo-granulated, and are considered to be adaptive MCs as a response to inflammation [17]. The differences concerning the functions depend on the response to different stimuli. 

The main role of tissue Mϕs is scavenging, i.e., recognizing and eliminating (by phagocytosis and silent degradation) endogenous threats such as senescent or transformed cells, dead cell, cellular debris, and misfolded/aggregated proteins. Both in physiological remodeling and in wound healing, Mϕs contribute to tissue re-shaping by their destructive capacity (mainly but not exclusively through the production of proteases and intracellular degradation in phagolysosomes; see the case of osteoclasts) and the parallel capacity of reconstructing tissues and promoting angiogenesis (via the release of growth factors such as FGF and VEGF, and via the production of matrix components) (Figure 2). Their specific activities are differentially triggered by microenvironmental cues (including changes in oxygen levels and temperature), to adequately react to diverse impending events [39]. Depending on the tissues in which they reside, Mϕs also display tissue-specific functions, as in the case of microglial cells, which function in the brain as sentinel cells in host defense, or splenic red-pulp Mϕs, which engulf erythrocytes, contributing to recycling heme [14]. An important example of the tissue-specific adaptation of Mϕs is represented by osteoclasts (OCs), large multinucleated cells that participate in physiological bone metabolism by resorbing bone in coordination with osteoblasts, which generate new bone [40]. Under stress conditions (such as infection or inflammation), tissue-resident Mϕs exploit their scavenging/phagocytic abilities for eliminating invading agents and, at the same time, they send alarm signals (e.g., alarmins and chemokines) that recall circulating leukocytes to the site of infection/inflammation. Blood-borne cells, including monocytes, are the major defensive effector cells that are able to kill invading agents by adopting the M1 killing phenotype in response to inflammatory environmental cues, with the production of inflammatory cytokines such as TNF-α, IL-1β, IL-6, IL-12, and IFN-γ, and toxic molecules such as reactive oxygen species (ROS), nitric oxide (NO), and antimicrobial peptides [33]. At the end of the inflammatory reaction, many tissue-resident Mϕs and effector monocytes have died, and the changes in the microenvironment drive the M2/healing polarization of the surviving Mϕs and the incoming monocytes. These cells contribute to extinguish inflammation (e.g., by producing anti-inflammatory mediators such as IL-10 and IL-1Ra) and to commencing the tissue reconstruction phase, using the same mechanisms that are employed for tissue remodeling; e.g., the release of growth factors and matrix components, and the replenishment of the pool of resident Mϕs by recalling new monocytes in a non-inflammatory manner (use of different chemokines, providing M2-polarizing factors) [31,32,41,42]. These circulating monocytes differentiate in tissue-resident Mϕ, which are sentinels that sense infection and tumor growth [11]. The M2-like functional phenotype is considered to be alternative or “anti-inflammatory”, is characterized by the increased production of IL-10, IL-4R, and VEGF, and the low expression of TNF-α and IL-12, promoting wound-healing, and in general, dampening Type 1 inflammation [43]. CD80, CD86, and CD64 are the most distinctive makers for human M1, CD163, and CD206 for M2 macrophage activation, and therefore, they can be helpful in determining the activation status of human macrophages [42,44,45] (Figure 2). However, the M1/M2 dichotomy represents the extremes of a plethora of cellular phenotypes with intermediate characteristics. In fact, there are many intermediate phenotypes (M2a, M2b, M2c, and M2d) that can produce both anti-inflammatory and inflammatory cytokines [32]. For example, the human placental populations of MΦs in preeclampsia are difficult to classify as M1 or M2, because they show characteristics of both phenotypes [46].

Human Mϕs are activated by TLRs with the aim of triggering the pro-inflammatory pathway and increasing the synthesis of antimicrobial peptides, cytokines, and chemokines to fight infections (Figure 2 and Table 1). 

Mϕs express the neurokinin-1 receptor (NK-1R), which triggered by SP induces inflammatory cytokines such as IL-6 and TNF-α, and increases oxidative stress through NO and ROS production, phagocytosis, and antigen presentation [38] (Figure 2 and Table 1). However, SP can also exert anti-inflammatory activity in LPS-stimulated Mϕs, promoting IL-10 and Arginase 1 (Arg1) expression, and inhibiting inflammatory cytokines production in mice [47]. Moreover, the M1- and M2-subtypes play important roles in allergic diseases [48] and can produce histamine [49], and lipid mediators as LKs and PGs. 

In general, in the case of mild infection or stress, Mϕs can resolve inflammation, but in case of strong infection or tissue damage, the recruitment of other specialized cells is essential, as is the case with neutrophils, for example. MCs cooperate with tissue-resident Mϕs to ensure the fast infiltration of neutrophils and their subsequent distribution over the affected tissue in mice [50]. Recruited cells contribute to sustaining and amplifying the Mϕs response, and the inflammation, if not resolved, can ultimately become chronic.

## 3. Physiological Role in Omentum

### 3.1. Peritoneum

The peritoneum represents a privileged site where both MCs and Mϕs are present under homeostatic conditions, and can be recruited to organs and tissues that are found in the omentum (uterus, ovaries, and placenta) to maintain the homeostasis of visceral organs. 

Peritoneal Mϕs (PMΦs) are of two subtypes: resident Mϕs with a high phagocytic capacity (named large peritoneal macrophages in mice (LpMs)) and monocyte-derived Mϕs recruited from blood (named monocyte-derived small peritoneal macrophages in mice (SpMs)) [51]. In homeostatic conditions, resident Mϕs represent most of the cells; they are longer-lived, exhibit self-renewal, and are derived from embryonic cells, having an anti-inflammatory phenotype [52,53]. In addition, chemoattracted short-lived monocytes (Ly6C^hi^) constitutively enter the peritoneal cavity in a CCR2-dependent manner (chemoattracted by CCL2 chemokine); they are induced to transiently differentiate into SpMs, and then into resident LpMs, and they can eventually displace embryonic-derived LpMs in mice with age [54]. Thus, the peritoneal population is constituted both by embryonic macrophages and by monocyte-derived macrophages, which have been shown to be transcriptionally distinct from each other [54]. The role of SpMs in homeostasis is unclear, but they are implicated in inflammatory reactions (see their inflammatory role). 

Peritoneal MCs (PMCs) are mature CTMC-subtype (MC_TC_) cells [55]. They represent <5% of cells recovered in homeostatic conditions. It was reported that PMCs contain higher amounts of histamine than other MCs in tissues, but less TNF-α. Moreover, FcɛRI aggregation triggers a release of high proteolytic activity, among them one selectively expressed in peritoneal cells hydrolyzing proteins such as casein and SH-2-containing inositol 5′ polyphosphatase 1 (SHIP1) [56]. Estrogen controls the homeostasis of many different tissues, including the peritoneum, uterus, and ovaries. It has been reported that in homeostatic conditions, rat PMCs that are challenged by estrogen did not give any detectable histamine release [57], whereas PMΦs were activated and expressed cell proliferation, immune response, and wound healing genes, showing a M2-phenotype signature [58]. The impact of estrogen levels and MΦ phenotyping is gaining intriguing insights, since it has been demonstrated that dysregulated levels of this hormone are implicated in some human pathologies such as endometriosis (see endometriosis).

### 3.2. Uterus

The human endometrium is a unique tissue that cyclically undergoes proliferation, secretory/differentiation, and shedding/repair because of estrogen and progesterone monthly fluctuations, during the menstrual cycle (Figure 3).

If fecundation does not occur, the upper functional layer of the endometrium breaks down and is shed at menstruation, showing the hallmarks of a pro-inflammatory physiological process [59]. The endometrial tissue that is adjacent to the myometrium (Figure 4) is involved in rapid re-epithelialization, the cessation of bleeding, and the regeneration of the functional layer. The nature of the endometrium is highly heterogeneous, comprising different cellular components, i.e., epithelial, stromal, vascular, and immune cells, each exhibiting a unique gene expression profile [60,61,62,63]. Among the immune cells there are neutrophils, uterine (u)Mɸs, eosinophils, basophils, uterine (u)MCs, and uterine natural killer cells, u(NK) [64]. All these immune cells contribute to the physiology of the reproductive system, each cooperating according to its role.

The number of uMϕs varies during the cycle, through their recruitment from blood circulation, and comprise 1–2% of endometrial cells in the proliferative phase, 3–5% in the secretory phase, and 6–15% in the menstrual phase [65]. UMCs are few, but their numbers and maturation strictly depend on estrogen and progesterone levels in mice [66], rats [67], and humans [68]. The presence of both MC_T_ and MC_TC_ subtypes in the human uterus, and in addition, in a minority of chymase-MC (MC_C_) subtypes, has been reported. It is now well established that these subtypes reflect different stages of differentiation or transdifferentiation in response to microenvironmental signals [68]. The MC_T_ subtype is abundant in the endometrium–myometrium during all stages of the uterine cycle, whereas the MC_TC_ subtype is found in all layers (Figure 4). 

#### 3.2.1. Proliferative Phase

During the proliferative phase, uMϕs express activation markers such as CD69 and CD71 and the adhesion marker CD54, suggesting their involvement in the proliferation and regeneration of endometrium tissue [69], and in gland remodeling by the phagocytosis of old secretory glandular epithelial elements [70]. UMϕs exhibits an alternatively activated phenotype (M2b-subtype), being able to produce both inflammatory (TNF-α and IL-1β) and anti-inflammatory (IL-10 and IL-1 receptor antagonist (IL-1RA)) cytokines, as well as high levels of angiogenic factors such as VEGF, FGF2, and PDGF [71] (Table 2). More recently, it was reported that estrogen can increase the proliferative capacity of Mϕs, with anti-inflammatory, and pro-resolving phenotypes [58]. 

It was reported that estrogen enhances the recruitment of MCs in the uterus [72] and increases their degranulation in vitro [73]. More recently De Leo and colleagues demonstrated a “recovery” state (steady state) during the proliferative phase for uMCs (endometrial and myometrial), suggesting that uMCs are not involved in this phase [68] (Table 2). 

**Table 2 ijms-23-05414-t002:** Cytokines and mediators produced by uterine mast cells and macrophages during the different phases of the menstrual cycle under physiological conditions. Abbreviations: uMɸs (uterine Mɸs); uMCs (uterine MCs); mMCs (myometrial MCs) [64,68,69,70,71,74,75,76,77,78].

Phase	Cell	Molecules/Status	Ref.
Proliferative	uMϕ	TNF-α; IL-1β; IL-10; IL-1RA; VEGF; FGF2; PDFG; phagocytosis; CD69; CD71; CD54	[69,70,71]
Proliferative	uMC	Steady state	[68]
Secretory	uMϕ	IL-11; activin; CD163 (70%)	[71,74,75,76]
Mid-late secretory	uMC	Tryptase; chymase; TNF-α; histamine; heparin	[68]
Menstrual repair	uMϕ	TNF-α; MMP-9; MMP-12; MMP-14 Phagocytosis; TIMP; VEGF	[64,77,78] [78]
Menstrual	mMC	Tryptase; chymase	[68]

#### 3.2.2. Secretory Phase

During the secretory phase, the levels of estrogen in the uterus and its receptor (ERs) in target cells decrease, while the levels of progesterone and its receptors (PRs) in the nuclei of epithelial and stromal cells increase, except in the case of MCs and Mϕs, which do not display the presence of the PR receptor [68]. The lack of PR expression in these cells supports the idea of indirect regulation, likely via factors that are secreted by progesterone-responsive cells. Endometrial cells [79] and perivascular cells express PR during the secretory phase, and are suggested to be the major targets of progesterone in the uterus and the cells that support the secretory phase [80].

Progesterone drives endometrial differentiation, known as “decidualization”, giving rise to cells that display a specialized secretory phenotype (Figure 3) [64]. Decidualization begins near the spiral arteries and initially only involves a few stromal cells, with the process then spreading throughout the endometrium, due to the paracrine action of cytokines such as IL-11 and activin (produced by stromal cells, resident MΦs, and neutrophils) (Table 2) (Figure 4) [74]. The decidualization process continues through the mid-secretory phase, which represents the implantation window for the trophoblast, if fertilization occurs. Chemokines that are present during the mid-secretory phase are likely to direct the recruitment/movement of leukocytes, and eventually the movement of the trophoblast. Both decidual endothelial and stromal cells express MCP-1, IL-8, IP-10, fractalkine (FKN), and stromal-derived factor (SDF)-1, and in addition, stromal cells express RANTES and MIP-1 [81]. All of these factors contribute to a local increase in the number of Mϕs, eosinophils, and neutrophils, along with a concomitant increase in macrophage-derived cytokines and proteases [82]. Many studies have reported that the majority of resident Mϕs (70%) showed anti-inflammatory phenotypes (CD163 expression), suggesting that the normal environment is anti-inflammatory, at least until menstruation, but that a small subset (approximately 30%) showed an inflammatory phenotype, and that the major percentage of uMϕs (~60%) is positive for TLR4 expression [71,76], suggesting that an aseptic environment is critical for ensuring trophoblast implantation. MCs during the secretory phase appeared to be activated in the myometrium, releasing both tryptase and chymase, although the stimuli responsible for their activation are largely unknown [68] (Table 2). In addition, uMCs can release TNF-α [83], which can induce the expression of activin in uMϕs, suggesting a crosstalk between them [84] and the production of proMMP-1 and -3 in uterine stromal cells. These pro-enzymes are activated by MC-tryptase and later by progesterone withdrawal [85,86]. All these biomolecules, including histamine [87] and heparin [88], participate in the remodeling of the uterine tissue, promoting vasodilation and edema, suggesting a strict collaboration between MCs, Mϕs, and stromal cells. Thanks to the presence of edema, many factors can be solubilized and made available in a regulated manner (spatial and temporal) for the target cells. A complex interaction between soluble factors released into the tissue by activated cells and the presence of specific receptors on target cells must be considered. 

If implantation occurs, the decidualization process continues (see placentation paragraph), otherwise, progesterone levels decrease (progesterone withdrawal), promoting a pro-inflammatory physiological microenvironment and the menstrual phase. 

#### 3.2.3. Menstrual Phase

Progesterone withdrawal is the trigger for menstruation (Figure 3). Among the late secretory and the pre-menstrual phase, a sequence of interdependent events of pro-inflammatory nature are initiated. ROS, which are physiologically produced by cells, accumulate because they are not eliminated; the activity of the superoxide dismutase (SOD) enzyme decreases in the glandular and stromal cells of the endometrium. The transcription factor NF-κB is activated by increased amount of ROS, and in turn, this drives the expression of numerous inflammatory chemokines and cytokines (CXCL8, CCL5, CCL1, CCL3, CXCL1, IL-6, IL-1β, and GM-CSF and TNF-α) as well as proteases (MMP-9) and enzymes (COX-2 and Mn-SOD) [64]. Indeed, COX-2 expression is rapidly elevated via NF-κB, leading to the production of PGs. PGs that are produced by stromal cells, and MCs and endothelins produced by stromal cells contribute to the vasoconstriction of arteries, inducing a hypoxic environment, which causes the expression and stabilization of the transcription factor HIF-1α (Figure 3) [89]. This latter condition promotes the expression of numerous target genes such as IL-8, CTGF, IL-1β, and TNF-α, including angiogenic factors in epithelial and stromal cells, which may later participate in endometrial repair during post-menses. Among the cytokines, IL-8, IL-1β, and TNF-α participate in the positive feedback loop to amplify the inflammation condition contributing to the selective recruitment of inflammatory cells (neutrophils, eosinophils, basophils, and monocytes) from the bloodstream into the endometrium [64] (Table 2). UNKs and MCs increase, partially through proliferation, but only MCs have a critical role in the menstrual phase. MCs appeared to be activated in the myometrial compartment (Figure 4) [68], although the stimuli for MC activation are unknown endothelins, which are produced by stromal cells, can activate MCs via its cognate receptor [90]. Crosstalk between MCs and Mϕs is likely to be involved in endometrial shedding and the breakdown of the uterine tissue (Figure 4). Indeed, MCs release tryptase, which plays an important role in activating proMMP-9, proMMP-12, and proMMP-14 released by resident Mϕs (Table 2). This breakdown has a focal nature and depends on the local activation of the latent enzymes (including proMMP). It is very likely that there is an activation cascade, including some proMMPs that are activated by MC-tryptase, and then some MMPs that activate other proMMPs. Furthermore, tryptase may play a role in degrading collagen VI that is present in the endometrial matrix [91]. Following the breakdown of the tissue, endometrial cells that are shed undergo apoptosis and are phagocytosed by monocyte-derived Mϕs that are recruited from blood, and which display an M2 phenotype [78] (Table 2). In addition to a scavenging role, M2 subtypes facilitate endometrial regeneration [92], as they play a role in angiogenesis and wound healing in other tissues [41] and can produce TIMP (which inhibits MMPs) and VEGF factor [64]. In addition, VEGF is produced by decidualized endometrial cells [93], and decidual VEGF plays a significant role in Mϕ recruitment and M2 polarization [94], which are detected transiently in the uterus and then undergo apoptosis when endometrial repair is complete [95]. Instead, resident Mϕ could be maintained by growth factors and cytokines, as with GM-CSF that is expressed in endometrial epithelial cells [96] and in uMC [97]. Interestingly, in areas of restoration, epithelial integrity were found in GM-CSF-polarized macrophages (M1-subtype) [98], suggesting a correlation between the polarization of MCs and Mϕs. In general, due to the local release of factors and cytokines and the focal nature of the simultaneous breakdown and repair process, it is difficult to exactly define the Mϕ phenotypes that are associated with each condition. Indeed, this field is still under study.

Therefore, it seems clear that MCs and Mϕs, together with endometrial stromal and epithelial cells, cooperate during the decidualization process and menstrual phase for the appropriate remodeling of the endometrium. Mϕs seem to depend on MCs for their pro-inflammatory polarization and functional activity.

#### 3.2.4. The Pregnant Uterus 

Successful pregnancy necessarily goes through two fundamental steps: implantation and placentation. Implantation of the blastocyst depends on extracellular matrix (ECM) digestion and remodeling of the spiral arteries in the endometrium, with the scavenging of apoptotic cells being generated by the invasion process of the trophoblast (i.e., the outer layer of a blastocyst). Therefore, at the fetal–maternal interface, there is contact between fetal (trophoblast) and maternal cells (endometrium), and they must communicate together for the formation of the placental bed and for the normal development of the fetus [99]. Indeed, the fetus is a semi-allograft and risks rejection if there is no fetal–maternal tolerance, which is reached thanks to fine regulation of the maternal immune microenvironment. When fecundation occurs, menstruation does not follow the secretory phase, and the decidualization process of the endometrial cells continues. Decidual Mϕs [100,101,102,103] and uMCs [104,105] can be found near the spiral arteries and the invading trophoblast, suggesting that these cells can prepare the endometrium for implantation. A pro-inflammatory microenvironment is linked to implantation, and endometrial vascular permeability is increased in the implantation site, which is mediated by COX-derived PGE_2_ [106] (Figure 5).

Bosquiazzo and colleagues conducted a study in rats in which they simultaneously analyzed the presence and activation of Mɸs and MCs [107]. The authors suggest that the two cell types may act in a coordinated manner because their numbers vary inversely during pregnancy, and Mɸs density is modulated by MC degranulation [107]. Eventually, implantation depends on the presence of uNK and uMɸs cells in the endometrium, which can produce both MMP-7 and MMP-9, which are involved in the degradation of the vascular ECM and spiral arteries, as well as in the apoptosis of endothelial cells [108] (Table 3). With regard to MCs, they produce histamine and chymase, and this latter contributes to the apoptosis of uterine smooth muscle cells, a key feature of the remodeling of spiral arteries [87,109,110] (Table 3). Therefore, Mɸs and MCs collaborate in inducing apoptosis of the cells that must be eliminated in the uterine tissue to favor the implantation of the blastocyst.

Decidual stromal cells increase the level of SCF expression, and SCF promotes mRNA expression of leukemia inhibitory factor (LIF) in decidual MCs [118]. LIF likely enhances trophoblast angiogenesis (tube formation) by decreasing the secretion of antiangiogenesis factor soluble fms-like tyrosine kinase-1 (sFlt-1) [119] that is produced by the M1-subtype, suggesting the ability of MCs to influence vascular remodeling by controlling the activation of Mϕs. 

After the early remodeling phase, Mɸs play a further role in the phagocytic clearance of apoptotic cells, preventing the release of inflammatory substances in the decidua [102,108] (Table 3). 

The glycan-binding protein galectin (Gal)-1 regulates multiple events that are associated with successful pregnancy, including trophoblast growth, syncytium formation, and angiogenesis in humans [120,121]. MCs produce and secrete Gal-1, which contributes to normal placentation through nonimmunological mechanisms, and support trophoblast survival [122]. 

Following implantation, an M2-like anti-inflammatory phenotype increases markedly, involving a high expression of IL-10 [111] with the purpose of maintaining fetal–maternal immune tolerance [36,99] (Table 3, Figure 5). 

The exact mechanisms for maintaining tolerance are far from being completely understood. Human leukocyte antigen-G (HLA-G), secreted by the trophoblast and later by the placenta, triggers cells into immunosuppressive phenotypes [123,124]. Decidual Mϕs do not express HLA-G, but they can respond to it because they possess the functional receptor, i.e., killer cell Ig-like receptor 2DL4 (KIR2DL4/CD158d) [125,126], as well as human MCs [127]. Decidual Mϕs that are exposed to HLA-G can produce IL-6, which inhibits the production of inflammatory cytokines by T lymphocytes facilitating fetal–maternal tolerance [115]. Moreover, they can produce PGE_2_, which can stimulate the synthesis of anti-inflammatory Th2 cytokines and suppress Th1 inflammatory cytokines [69] (Table 3). On the other hand, HLA-G binding to its receptor on MCs induces the production of LIF and serine proteases. LIF increases STAT3 activation and, as mentioned above, trophoblast migration and angiogenesis (tube formation) by decreasing sFlt-1 secretion, while serine proteases degrade protease-activated receptors (PARs) and help to promote trophoblast tube formation [119]. It has been reported that the HLA-G-KIR2DL4 axis may play an important role in promoting pregnancy by suppressing the cytotoxic activity of decidual NKs [119]. 

In humans, MCs increase in the uterine cervix during pregnancy, promoting cervical ripening [128]. The role of MCs in the establishment of pregnancy is not yet clear and this is currently under investigation. However, histamine that is released from decidual MCs is involved in blastocyst implantation, and the invasion/proliferation of villi [110]. Chymase is required for decidual vascular remodeling and fetal growth, having a role in maintaining steady blood supplies for the fetus [110], whereas tryptase and leptin [129] are involved in angiogenesis, together with VEGF and FGF in neovascularization (Table 3). 

So far, studies that have been conducted in the early stages of human pregnancy seem to confirm a collaboration between MCs and Mϕs for the success of implantation; for example, in tissue remodeling and maternal–fetal tolerance, which are objectives that are achieved with the production of different biomolecules.

### 3.3. Placenta

Human placental Mϕs are composed of two distinct populations, i.e., decidual Mϕs (maternal) and Hofbauer cells (fetal), these latter have been suggested to be of M2-like phenotype [130]. During pregnancy, the number of decidual Mϕs varies, reaching the most significant amount during the first and second trimester, which is also due to their recruitment via specific chemokines. Indeed, decidual Mϕs comprise approximately 20–30% of all human decidual leukocytes in early pregnancy, and thereafter, the number decreases [131]. An imbalance in immune cell proportion in the placental bed can lead to diseases and pregnancy complications.

The origin of the Hofbauer cells is still debated. According to some authors, they derive from mesenchymal cells within the stroma of the chorionic villi [132,133]; according to others, they derive from progenitor monocytes in the yolk sac of the embryo [134,135]. Instead, decidual MΦs derive from maternal HSCs that differentiate into circulating monocyte progenitors, passing into the placenta, where they mature. They can self-renew, thanks to placental overexpression of IL-4 and M-CSF, which also can induce their polarization [136]. In addition, IL-34 that is produced by maternal decidua contributes to the anti-inflammatory polarization of Mϕs, while GM-CSF of fetal origin (trophoblast) induces pro-inflammatory polarization [137,138,139,140].

Therefore, placental MΦs are a heterogeneous cell population of fetal and maternal hematopoietic origin, and these latter can change their phenotype with gestational age (Table 4). More precisely, some studies have highlighted the presence of two different decidual Mϕs subsets, a majority (70%) with anti-inflammatory and a minority (30%) with pro-inflammatory characteristics [36,46] (Table 4). Instead, other studies have highlighted the presence of two subtypes with the ability to produce both inflammatory and anti-inflammatory cytokines, suggesting that they do not fit conventional M1/M2 categorizations [99,103] (Table 4). 

However, single-cell transcriptome analysis highlights a high degree of cellular heterogeneity and different maternal cellular subtypes in the human placenta [143]. More recently, Xiangxiang Jiang et al. found in the decidua three subsets of MΦs [144]. The more numerically consistent subset (CD11c^low^ 80%) was found to be scattered in the decidua and it has immunosuppressive characteristics; the other two phenotypes (CD11c^high^ 5% and 15%) were mainly detected near the extravillous trophoblast cells (the cells that infiltrate the maternal decidua) with pro-inflammatory or antioxidant and anti-inflammatory characteristics, respectively, as well as displaying different phagocytic abilities [144]. Their simultaneous presence in the decidua of pro-inflammatory and anti-inflammatory Mϕs phenotypes, during active gestation, is related to their role as immune cells for facilitating the clearance of pathogen infection, as well as for the maintenance of homeostasis at the maternal–fetal interface, an ability that Mϕs share together with MCs. MCs are rare in the placenta, and their numbers, as well as their activation status, are strictly controlled. There are no studies that directly compare MCs with Mϕs in the placenta, while Tregs, uNK, and decidual stromal cells (DSCs) are found as possible partners. A recent study has shown that under homeostatic conditions, MΦs are induced by DSCs to polarize to the anti-inflammatory phenotype (M2) [145]. Another study showed that placental MCs interact with decidual Tregs; these latter produce IL-9, which inhibits the IgE-mediated activation/degranulation of MCs but induces the release of anti-inflammatory cytokines (TGF-β and IL-10), which supports the implementation of maternal–fetal tolerance [146]. In a study conducted early in the first trimester of pregnancy, woman who had had a spontaneous abortion had increased numbers of decidual MCs, suggesting a negative correlation with a successful pregnancy [147], and their pattern was the MC_TC_ subtype, showing granules with a higher histamine content [117]. These studies suggest that MCs are in a resting state during pregnancy and release a physiological level of mediators that maintain homeostasis, essentially contributing to fetal–maternal tolerance. Any change in their state of activation induces an imbalance of the signals in the uterine microenvironment, with possible consequences on pregnancy (see the paragraph on preeclampsia). Therefore, since MCs appear to be in at a steady-state level during early pregnancy and likely contribute to a physiological release of their mediators, the interaction between MCs and Mɸs in this phase is limited.

In homeostatic conditions, the pregnancy (active gestation) is wholly associated with an anti-inflammatory environment (Figure 5). This possible explanation lies with steroid hormonal control of the cellular activation status, since progesterone and estradiol levels increase during pregnancy. Unexpectedly, it was reported that progesterone and estradiol have no direct effects on the uterine Mϕ phenotypes, while an indirect effect, i.e., mediation by cells that are responsive to hormones, cannot be excluded. For example, rat uterine MCs are activated by estradiol and can release mediators [73,97], which may influence Mϕ polarization. During full-term pregnancy, and more precisely, at labor, the scenario changes profoundly. In fact, the microenvironment typically becomes inflammatory and the number of decidual M1-subtypes and MCs in cervical biopsies increases, suggesting that a pro-inflammatory phenotype may play an important role in the onset of term labor (Table 4 and Figure 5) [141,142].

Overall, under physiological conditions, the collaboration between MC and Mϕ after implantation is mainly aimed at maintaining maternal–fetal tolerance, and promotes term labor. However, this collaboration is likely not direct, since from the studies conducted to date, the interaction appears to be mediated by other cellular partners or biomolecules that are produced by other cells. In addition to these functions, the MC and Mϕ may play other exclusive roles that are all fundamental for physiological pregnancy.

## 4. Inflammatory Roles in the Omentum

### 4.1. Peritoneum

Under inflammatory conditions, LpMs migrate into the omentum and their numbers in the peritoneal cavity decrease. Those cells that remain in the peritoneum can play a regulatory role in dampening inflammation by producing IL-10 cytokine, and by regulating the number of inflammatory SpMs [51]. Concurrent with the reduction in the number of LpMs, an increase in both the number of pro-inflammatory monocytes recruited and SpMs is observed with the ability to produce inflammatory cytokines. When inflammation is resolved, the pro-inflammatory subtype will then die, and the LpMs-depleted population restores its numbers through the proliferation of CSFR-1-mediated cells exhibiting a pro-repair phenotype [148]. This is a virtuous circle, with the inflammation at the beginning being the predominant response, but with later resolution and tissue repair responses expected to prevail [149]. Therefore, polarized Mϕs are still able to reversibly undergo functional re-differentiation into anti-inflammatory or pro-inflammatory phenotypes [53]. 

In acute sepsis, the functional crosstalk between peritoneal MCs and Mϕs has detrimental effects on mouse survival. Indeed, MCs impair the phagocytic action of resident Mϕs through the release of pre-stored IL-4 a few minutes after bacteria exposition [150]. In peritonitis, peritoneal Mϕs and MCs increase in COX-1 and COX-2 activities and produce the chemokines PG [151] and MMP-9 [152]; this latter is critical for neutrophil infiltration into the peritoneum [152]. Ultimately, MCs play a role in the initiation of inflammation, but many studies indicate that the main contribution of MCs is in the modulation of the pro-inflammatory response, whereas Mϕs are critical for its resolution [153]. Indeed, experiments of co-culture between PMCs and PMϕs have shown that activated PMC IL-33 promotes the anti-inflammatory polarization of PMϕs by secreting IL-13 and IL-6 [154]. In another study, bone marrow (BM)MCs co-cultured with *C. albicans* produced TNF-α, IL-6, IL-13, and IL-4, and increased the crawling ability of PMϕs to the site of infection. On the other hand, resting MCs inhibited phagocytosis by Mϕs versus *C. albicans* in a contact-dependent manner [155]. MCs triggered, for example by compound 48/80, released inflammatory mediators such as TNF-α, IL-6, histamine, LKs, PGs, and proteases, and resulted in PMϕ activation [156]. In turn, activated Mϕs can produce TNF-α, switch on enzymes such as iNOS and COX-1, and further amplify inflammation [157], then they can activate COX-2 enzyme and produce IL-10 to resolve the inflammation. These studies suggest the ability of MCs to control PMϕ polarization under non-physiological conditions, and to restrain their effector functions under homeostatic conditions. 

Overall, the signals present in the peritoneal environment during inflammation determine the activation pattern of Mϕ and condition their responses, which is decisive for the outcome of the inflammatory response. The role of MC is to orchestrate the activation of Mϕs and their polarization, guaranteeing the pro-inflammatory response and subsequently the successful outcome of the resolution.

### 4.2. Uterus

#### 4.2.1. Endometriosis

Endometriosis is a debilitating chronic inflammatory disease affecting women of reproductive age, and it is associated with the growth and proliferation of endometrial-like tissues outside the uterus (ectopic endometrium). The main consequences of this disease are chronic pelvic pain and/or infertility, which is due to the uterine endometrium (eutopic endometrium) being unsuitable for embryo implantation [158]. In fact, the eutopic endometrium exhibits pro-inflammatory transcriptomic features [159], and despite elevated levels of estrogen, is unable to elicit a normal hormonal response [160,161]. uMCs and uMΦs participate in the pathogenesis of the disease as well as PMCs and PMΦs, but their specific role is still under study [76,116,162,163,164,165,166]. Hogg and colleagues demonstrated that MΦs, in ectopic lesions, are derived from eutopic endometrial tissue from recruited peritoneal LpMs and monocyte-SpMs [51]. The local environment of the peritoneal fluid (PF) surrounding the endometrial tissue is dynamic and contains a variety of immunologically cells, chemokines, and growth factors, including SCF [167] and M-CSF [168], which regulate the differentiation of MCs and MΦs, respectively. Among the chemokines, CCL2 (encoded MCP-1) is abundant in the PF and in the lesions of patients with endometriosis, and its cognate receptor (CCR2) is upregulated in monocytes. Therefore, blood monocytes are continuously chemoattracted into the peritoneal cavity (via CCL2–CCR2) [54] to replenish the number of monocyte-SpMs that are in turn recruited into the lesions (via CCL2 over-expression in stromal cells with an estrogen receptor-dependent mechanism, (ER)-β) [169]. It was proposed that the resident MΦs in the lesions are pro-endometriosis (M1-subtype), while the monocytes recruited from the peritoneum are anti-endometriosis (M2-subtype) [51]. 

In endometrial lesions, the number of MCs increases and these are located around blood vessels and close to nerve fibers [170]. SP, a neuropeptide with the function of pain mediation in endometriosis [171], triggers MCs, which degranulate, releasing TNF-α and IL-6 [24]. Previous studies have demonstrated that in endometriotic lesions, the number and activation of MCs are associated with the levels of serum estrogen, TNF-α, and nerve growth factor (NGF) [172], and the expression of CCL8 chemokine [173]. The CCL8–CCR1 axis promotes the migration/proliferation of both endometrial, epithelial, and stromal cells in the lesions, since the cognate receptor (CCR1) is up-regulated in these cells [173]. Recently, a study revealed that estrogen promoted the expression and activation of the NLRP3 inflammasome signaling pathway in MCs, via the estrogen receptor (ER)-α. This activation increased the MC production of inflammatory IL-1β cytokine and the overexpression of IL1R1 (IL-1β receptor) in the ectopic endometrial stroma [174]. The activation of the IL-1β-IL1R1 axis in stromal cells favored the production of IL-33, and this latter induced the polarization of recruited peritoneal monocytes towards the M2-subtype. In turn, the M2-subtype began to produce IL-1β which, as positive feedback, induced stromal cells to produce IL-33 in the lesions [175]. It is now widely known that the development of endometriotic lesions is associated with the presence of the M2-subtype and the IL-33 cytokine [175,176], since the phagocytic ability of the M2-subtype (phagocytosis of the peritoneal ectopic endometrium) is decreased by the effect of IL-33 [48]. 

To date, studies suggest that MC and Mϕs have different roles in this disease, as activated MCs may play a role in promoting an inflammatory microenvironment and neurogenic pain, although their contribution to Mϕ activation/polarization directly or indirectly cannot be excluded. On the other hand, Mϕs can support inflammation underlying endometriosis through their phenotypic polarization in response to microenvironmental signals.

#### 4.2.2. Preeclampsia

Preeclampsia is the major placenta-related inflammatory disease of pregnancy. It includes two subtypes, based on the temporal (early and late) onsets of the disease, which have distinct pathophysiological origins. Indeed, early preeclampsia is mainly due to decreased trophoblast invasion and spiral artery remodeling [177], whereas late preeclampsia is caused by an imbalance in circulating antiangiogenic and angiogenic factors. However, both subtypes involve maternal and placental immune system dysregulation. So far, studies that have been conducted on the placentas of patients with preeclampsia have not given definitive answers on the roles of MCs and Mɸs in this disease, nor on the relationships between them. Additionally, there are contradictory results on the number of cells present in the placentas of patients compared to those of healthy women [102], while instead, the presence of an inflammatory microenvironment has been demonstrated. Among the studies that found an increase in the number of MCs, many also found that these cells were activated and degranulated by releasing histamine [178,179], tryptase, SCF, and MMP-2 [63] into the microenvironment. The high histamine levels released elicited a pro-inflammatory response with the secretion of cytokines and molecules, contributing to an increase in blood pressure, which is typical of preeclampsia [180]. Some studies reported fewer Mϕs in the preeclamptic placenta, whereas others reported an increased number, mainly localized near the uterine spiral arteries; the local accumulation probably being due to the recruitment chemotactic factors [102]. However, the status of Mϕ activation in ill and healthy subjects is profoundly different. Indeed, more of the M1-subtype and less of the M2-subtype are found in the preeclamptic placental bed, in agreement with more pro-inflammatory and fewer anti-inflammatory cytokines levels found in the placentas of patients [102,181]. In agreement, a recent study highlights an increase in the M1-subtype signature with the augmented production of pro-inflammatory cytokines such as IL-1β, IL-6, and chemokine CCL2 in the serum and in the human placenta of women with preeclampsia [182]. Another clue in favor of an inflammatory environment is given by the presence of high amounts of TLR4 in the vascular endothelium of the placental villi [183] and in circulating monocytes of woman with early-onset preeclampsia [184], suggesting a pro-inflammatory phenotype. Moreover, in preeclamptic women, observations have reported an increase in the production of free radicals and ROS, and in turn, oxidative stress, which leads or contributes to inflammation and preeclampsia [184]. As mentioned above, a cause of early preeclampsia is defective angiogenesis and the remodeling of the spiral arteries. Inflammatory macrophages (M1) are known to secrete an anti-angiogenic molecule, sFlt-1, suggesting that sFlt-1 may contribute to impaired angiogenesis in preeclampsia [181].

Other studies have investigated cellular signatures using high-throughput methods (transcriptomics and proteomics), which are not informative for the pathophysiology of preeclampsia, since they do not clarify either the single contributions of cellular subtypes or if the detected changes (number, activation, or alteration in phenotype) are causes or effects of the disease. However, even in these studies, the up-regulation of both the immune response and the gene expression of pro-inflammatory cytokines, including IL-6, IL-8, and TNF-α [185,186], and VSIG4 (V-set and immunoglobulin domain containing 4) are highlighted [37]. 

To the best of my knowledge, there is only one study in the literature that analyzes MCs and MΦs together in this disease. Broekhuizen and colleagues analyzed the placentas of patients with early-onset preeclampsia and found a reduced number of MCs and M2-subtypes compared to healthy placentas [187]. The decreased number of MCs in the villi of early preeclamptic placentas agrees with previous research [188], and their depletion results in aberrant spiral artery remodeling and eventual fetal growth restriction [110], since MCs are a source of angiogenic factors. In addition, the M2-subtype found within the placental villi has been found to decrease, and this loss could be related to a reduced clearance of apoptotic bodies and the insufficient protection of the fetus against intrauterine infection [114,189].

Overall, MCs appear to participate in the systemic responses observed in women with preeclampsia, through the production of histamine, which increases blood pressure, although it is not possible to exclude their role in inflammation events that develop both systemically and in the placenta. Instead, Mϕs play a role in promoting and supporting the inflammatory microenvironment, as demonstrated by the fact that a reduced number of the M2-subtype is already present in the first trimester of pregnancy, and is linked with fetal growth restriction and pregnancy-induced hypertension [190]. To date, there are no data in favor of a crosstalk between MCs and Mϕs in the context of this disease, but this cannot be excluded, since under homeostatic conditions, they collaborate for a successful implantation.

## 5. Conclusions

Inflammatory responses are critical for host defense and immunity, and the ability to restore homeostasis is essential for maintaining tissue integrity and function. Mϕs are essential actors for inflammatory events because they can possess two main roles: initiating and supporting inflammation, and restoring the conditions of homeostasis. On the other hand, MCs are also protagonists of inflammation, as they can induce the polarization of Mϕs in some contexts, and therefore change the course of events. Moreover, MCs have more complexity than has previously been thought, since they can possess a kind of polarization as well. To complicating the scenario, it must be considered that in addition to the cells residing in particular tissues, there are cells that are recruited from other districts. The interaction between tissue-resident and recruited cells is now at the forefront of research, due to their possibly different roles in inflammatory diseases. Much progress has been made in inflammation research, but there are still a few works that highlight the interactions between MCs and Mϕs. We must consider that both of these innate immune cells are in the same place and at the same time, and it is therefore obvious to assume that they must interact and influence each other. Furthermore, both cell types are part of an organism’s innate immunity; therefore, they are the body’s first line of defense against infection. For all of these reasons, further studies are needed to investigate the interactions of these cells within the same context.

## 6. Final Remarks

In the literature, there are very few works that study the presence and reciprocal interactions of MCs and Mϕs in different tissues of mammals. This review arises from the need to jointly analyze the activities of these cells under some physiological and pathological conditions related to pregnancy. It is my opinion that the simultaneous presence of MCs and Mϕs in the uterus, placenta, and peritoneum support an existing collaboration for pregnancy success. Sometimes it seems that the two cell types perform partially overlapping or very similar activities, but with a careful reading of microenvironmental signals, times, spaces, or the presence/activity of other leader cells, the absolutely necessity for the presence of both cells in humans can be understood. To the best of my knowledge, this paper is the first to review the crosstalk/interactions between mast cells and macrophages during pregnancy.

## Figures and Tables

**Figure 1 ijms-23-05414-f001:**
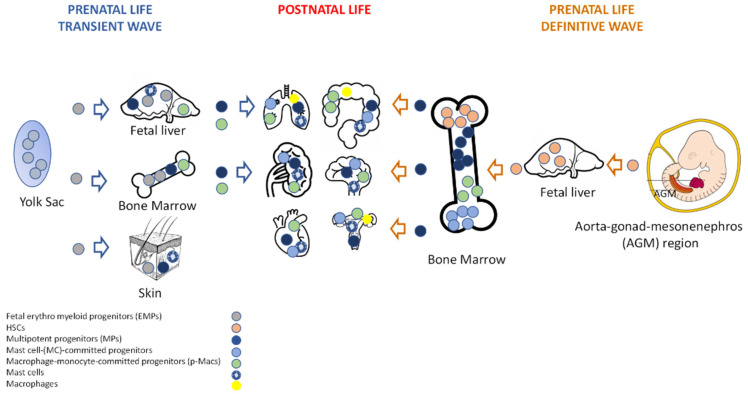
Schematic view of fetal hematopoiesis. Transient (definitive) and definitive waves, as well as local organ distribution of multipotent progenitors, committed-progenitors, and mature cells are shown. These sequential waves partially overlap in time and space. This figure was generated with BioRender and PowerPoint.

**Figure 2 ijms-23-05414-f002:**
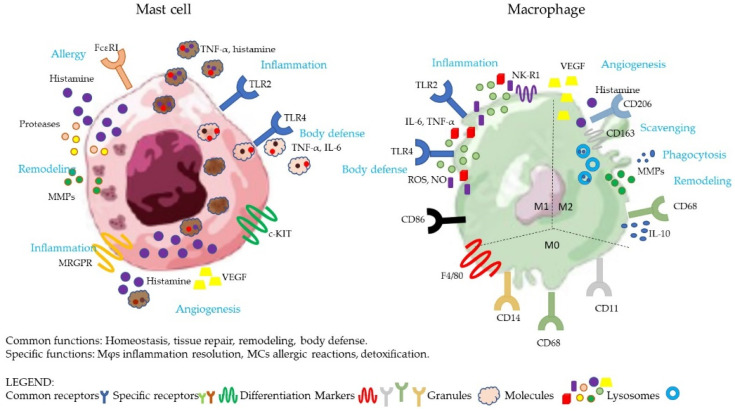
Representative picture of the main characteristics of MCs and Mϕs listed in this review. The mediators, cytokines, and enzymes produced following the activation of the receptors present on the cells are shown in the figure and are described in the text of the review. This figure was generated with Adobe Illustrator and PowerPoint.

**Figure 3 ijms-23-05414-f003:**
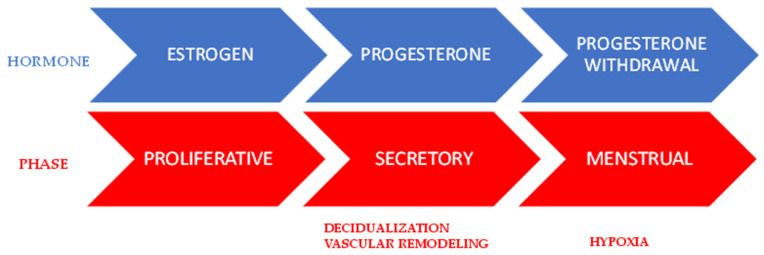
Timeline of hormone production and different phases during the menstrual cycle in humans. Figure generated using PowerPoint.

**Figure 4 ijms-23-05414-f004:**
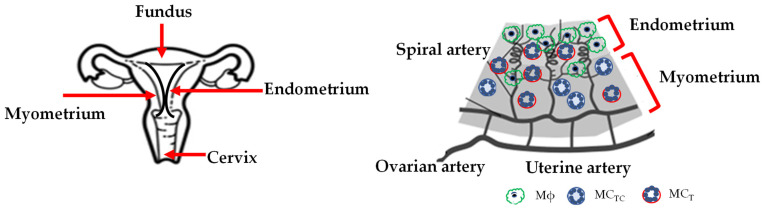
Schematic picture of the cycling non-pregnant human uterus with endometrium and myometrium tissue, and Mϕs, MC_T_, and MC_TC_ localization under physiological conditions. Abbreviations: Mast cells tryptase (MC_T_), Mast cells tryptase, chymase (MC_TC_). This figure was generated using BioRender and modified in PowerPoint.

**Figure 5 ijms-23-05414-f005:**
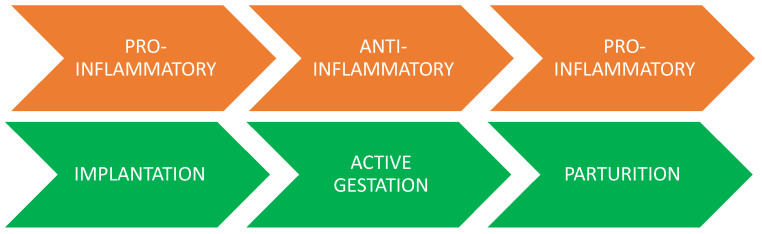
Timeline of pregnancy steps and the pro-inflammatory and anti-inflammatory physiological conditions of the microenvironment generated by the activity of immune cells. Figure generated with PowerPoint.

**Table 1 ijms-23-05414-t001:** Schematic description of the MC and Mɸ receptors, ligands, and functions mentioned in this review.

Receptor	Ligand	Expression	Function
TLR2	Bacterial, viral, fungal and parasites lipids, proteins, polysaccharides	MC and M1 M2	Pro-inflammatory cytokines production, degranulation [28,29,30], IL-10, 15-LO [30]
TLR4	Bacterial and viral proteins	MC and M1 M2	Pro-inflammatory cytokines production, degranulation [25,31,32,33,34], IL-10, 15-LO, 15-HETE lipoxins [34]
FcεRI	Allergens	MC	Allergic reaction [13,17,35]
CD14	Endotoxin receptor	Mɸ	Pro-inflammatory cytokine production [36,37]
F4/80	Marker mouse	Mɸ	Differentiation, induction of CD8+ T regulatory cells [10]
c-KIT	SCF	MCs	Migration, differentiation, and survival [12,13,21]
NK-R1	Substance P	Mɸ	Pro-inflammatory cytokines, NO, ROS, production. Induction of COX-2 activity [38]
MRGPR	Substance P	MC	Pro-inflammatory cytokine production, histamine and protease release [23]

**Table 3 ijms-23-05414-t003:** MCs and Mϕs phenotypes, functions, and biomolecules produced during different stages of early pregnancy. Abbreviations: MC_C_—mast cell chymase, MC_TC_—mast cell tryptase-chymase [36,63,69,72,73,87,97,99,102,107,108,109,110,111,112,113,114,115,116,117].

Time	Phenotype	Function	Production	References
Pre-implantation/human, mouse	M1	ECM digestion, engulf apoptotic cells	MMP-7; MMP-9; Gelatinase	[102,108,111,112]
Pre-implantation	M2	Remodeling	VEGF, IGF, Fn1, MMPs	[102]
Pre-implantation/human, murine	MCs	Remodeling angiogenesis trophoblast growth, syncytium formation	Histamine, chymase, VEGF, Gal-1	[72,87,107,109,110]
Implantation	M1	Favor implantation, vascular remodeling, angiogenesis	TNF-α, VEGF, PDGF, FGF, MMP3, MMP9; IL-1β	[111]
Implantation/human, mouse	MC_C_	Vascular remodeling, fetal growth, migration and proliferation of extravillous	MMPs, VEGF, FGF, histamine, tryptase, leptin, cytokines	[63,73,97,110,113]
Post-implantation	M2	Maternal–fetal tolerance	IL-10, IL-1β, Il-6, TNF-α; PGE2	[36,69,99,103,112,114,115]
Post-implantation	MC_TC_	Maternal– fetal tolerance	TGF-β IL-10, LIF	[116]
Early pregnancy	MC_TC_	Placenta formation	Histamine, Gal-1	[117]

**Table 4 ijms-23-05414-t004:** Placental MCs and Mϕs: Appearance, function, and biomolecules produced during different stages of pregnancy. Abbreviations: MC_C_—mast cell chymase, MC_TC_—mast cell tryptase-chymase [36,37,141,142].

Time	Phenotype	Function	Production	References
1st–2nd trimester	M1	Maternal–fetal tolerance	TNF-α, IL-6, IL-12, IL-23, IFN-γ, IL-18	[37]
1st–2nd trimester	M2	Maternal–fetal tolerance	IL-4, IL-10, VEGF, IGF, FN	[36]
2nd–3rd trimester	M2	Maternal–fetal tolerance	VEGF, IL-6, IL-10	[36]
End of pregnancy	M1	Favor labor, remove apoptotic cells	IL-6, IL-10, IL-1, TNF-α, IFN-γ	[36,37,141]
End of pregnancy	MC	Favor labor	Histamine	[142]

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
