# Peer review of "Interactions between Macrophages and Mast Cells in the Female Reproductive System"

_ijms, 2022, doi:10.3390/ijms23105414_

Round 1
Reviewer 1 Report
This manuscript aims to review the literatures on macrophage and mast cell interaction during homeostatic and pathological conditions. While the topic is interesting and important, the informativeness of the article is diminished by limitations in the organization, specificity and nuanced discussion. In addition, there are several writing errors that need to be addressed. Major revisions in these areas, as described below, would greatly improve the manuscript.
- Several sections in this review contains largely overlapping information. For example, the authors first discussed the plasticity of macrophage in section 1 “development and tissue distribution” (line 36-42). This discussion appears again in section 2 “biological function (line 176-205)” with redundant and repetitive information.
- Although the review is titled “macrophage and mast cell interaction”, there is actually little content regarding the crosstalk and mutual effects between these two innate immune subsets. The author simply listed the basic literatures of these two cells and barely synthesize any insightful information.
- Several elements highlighted and elaborated by the authors seems to be largely irrelevant to the main idea of the paper. For example, the authors use entire figure 3 to illustrate hormone production and phases of human menstrual cycle, and figure 4 to describe the anatomy of uterus, without mentioning how is this important for the discussion of macrophage and mast cell interaction. Consolidating those figures and tables and adding some kind of connection will make more sense.
Author Response
Reviewer 1
1. Several sections in this review contains largely overlapping information. For example, the authors first discussed the plasticity of macrophage in section 1 “development and tissue distribution” (line 36-42). This discussion appears again in section 2 “biological function (line 176-205)” with redundant and repetitive information.
A) The first two paragraphs have been revised with the aim of eliminating all repetitive and / or unnecessary information. However, regarding the plasticity of macrophages, it should be noted that in the first paragraph (line 36-42) there is only a short and general hint, whereas in the second paragraph (line 176-205) the characteristics of the subtypes (M1 and M2) are described in detail. It is my opinion that the information is necessary and not redundant.
2. Although the review is titled “macrophage and mast cell interaction”, there is actually little content regarding the crosstalk and mutual effects between these two innate immune subsets. The author simply listed the basic literatures of these two cells and barely synthesize any insightful information.
A) I totally agree with the reviewer regarding the paucity of information on crosstalk between the two cell types in the literature, and indeed this is written in several parts of the review. Not only there is a lack of studies comparing the two cell types in the literature, but there is also a lack of studies that analyze the two cell types simultaneously and in the same tissue. Nevertheless, all the papers published on the subject, even if not recent, have been analyzed and summarized in this review. Where possible, links, quotes, references and tips have been provided to make this review useful for readers. Furthermore, the conclusions underline the need for further studies on the subject, with the aim of analyzing the two cell types simultaneously with precision to define their possible crosstalk.
3. Several elements highlighted and elaborated by the authors seems to be largely irrelevant to the main idea of the paper. For example, the authors use entire figure 3 to illustrate hormone production and phases of human menstrual cycle, and figure 4 to describe the anatomy of uterus, without mentioning how is this important for the discussion of macrophage and mast cell interaction. Consolidating those figures and tables and adding some kind of connection will make more sense.
A) Figures 3 and 4 have been added to better visualize the phases of the menstrual cycle and the cyclical nature of the hormones, as well as the anatomy of the uterus (albeit in an extremely schematic form). In this way, it is possible to widen the audience of readers also to those who are not a reproductive immunologist. However, as suggested by the referee, the importance of hormones and anatomy of uterus, as well as menstrual cycle, were better emphasized in the text.
Reviewer 2 Report
The manuscript of Lampiasi is a high-quality, detailed review dedicated to the roles on MPhs and MCs and their interaction in physiological and pathological events in the female reproductive system. It is worth noting that despite the emerging evidences on key roles of immunity in pregnancy and foetus development, the number of research works in the area remains insufficient. Thus, the current review can add essential knowledge in how immune system is participating in the reproductive function and relevant events.
Minor comments
- It is recommended to include more detailed figure legends. It would improve the understanding of the manuscript.
- If possible, please add the MPh and MC cell subsets on Fig. 4 according to their localization.
-
Please specify the digestion enzymes (Table 3). It would be more logically correct to change the orders of column 1 and 2 (Tables 3-4), since the tables are organized according to the pregnancy stages.
Major comments
1. I would recommend to change the title to more associated with the roles of MPhs and MCs to the female reproductive system. The adopted title can more accurately reflect the paper's content and attract the researchers in immunology and reproductive biology.
2. Furthermore, more attention to the reproductive systems can be highlighted in the text. Ex., "3. Physiological role" can be changed to "3. Physiological role in omentum", and so on. The first part's title can be modified to General biological functions. The whole first part can be shortened. In the first paragraph the origin of the MPhs and MCs of the reproductive system can be separated from the general story of immune cell origin. The subtitles of p.3 (3.1-3.4) can be preferably referred either to location (ex., 3.4 Placenta) or to process (ex., 3.3 Implantation) but not mixed together.
3. The idea of Fig. 5 is not very clear. Historically, inflammation is more relevant to the disease pathogenesis rather than to the normal physiological events. Possibly, it is better to refer to the immune cell functionality in the mentioned stages (ex., proinflammatory, etc.). Similar question is rising for p.4 - whether this is pathological inflammation or normal proinflammatory activities of immune cells in unclear.
4. In Fig. 1: please note that hematopoiesis occurs simultaneously in bone marrow together with the yolk sac during the earliest developmental stages.
Overall, the manuscript is a good quality in terms of the scientific content, structural organization, literature sources. According to reviewer's knowledge this manuscript is among the first review article, which discuss the interactions of MPhs and MCs in reproductology. This review will bring substantial improvement in systematization and understanding of the immune cell behavior in female reproductive system.
Author Response
I am very grateful to the reviewers for their suggestions, which improved the quality of the manuscript.
Minor comments
1. It is recommended to include more detailed figure legends. It would improve the understanding of the manuscript.
A) As suggested by the reviewer, figure legends have been improved.
2. If possible, please add the MPh and MC cell subsets on Fig. 4 according to their localization.
A) As suggested by the reviewer, mast cells subtype and macrophages have been added in the new Figure 4.
3. Please specify the digestion enzymes (Table 3). It would be more logically correct to change the orders of column 1 and 2 (Tables 3-4), since the tables are organized according to the pregnancy stages.
A) As suggested by the reviewer, digestion enzyme (gelatinase) has been added in Figure 3, and Tables 3-4 have been organized as suggested.
Major comments
1. I would recommend to change the title to more associated with the roles of MPhs and MCs to the female reproductive system. The adopted title can more accurately reflect the paper's content and attract the researchers in immunology and reproductive biology.
A) As suggested by the reviewer, the title has been changed.
2. Furthermore, more attention to the reproductive systems can be highlighted in the text. Ex., "3. Physiological role" can be changed to "3. Physiological role in omentum", and so on. The first part's title can be modified to General biological functions. The whole first part can be shortened. In the first paragraph the origin of the MPhs and MCs of the reproductive system can be separated from the general story of immune cell origin. The subtitles of p.3 (3.1-3.4) can be preferably referred either to location (ex., 3.4 Placenta) or to process (ex., 3.3 Implantation) but not mixed together.
A) All suggested changes have been made. The first and second paragraphs have been shortened. As for the subtitles of paragraph 3, I preferred to insert "Implant" in the paragraph "Uterus" (new 3.2.4) and change the title in “uterus pregnant”. As for the first paragraph, the description of the origin of the two cell types is general. The details on mast cells and macrophages ontogenesis in the reproductive system are provided in the dedicated paragraphs (example: peritonum, uterus, placenta).
3. The idea of Fig. 5 is not very clear. Historically, inflammation is more relevant to the disease pathogenesis rather than to the normal physiological events. Possibly, it is better to refer to the immune cell functionality in the mentioned stages (ex., proinflammatory, etc.). Similar question is rising for p.4 whether this is pathological inflammation or normal proinflammatory activities of immune cells in unclear.
A) As suggested by the reviewer, it has been better clarified that the process concerns a physiological inflammation in Figures 4 and 5 (figure legends and text).
4. In Fig. 1: please note that hematopoiesis occurs simultaneously in bone marrow together with the yolk sac during the earliest developmental stages.
A) I'm not sure I understand correctly. However, I added a detail in the text "Foetal hematopoiesis occurs in three sequential waves that differ in time and space but partially overlap: the primitive, the transient (definitive), and the definitive wave", and in the figure legend “Schematic view of foetal hematopoiesis. Transient (definitive), and definitive wave as well as local organs distribution of multipotent progenitors, committed-progenitors and mature cells is showed. These sequential waves partially overlap in time and space”.
Round 2
Reviewer 1 Report
Thanks for the authors response. Modifying the title to to accommodate the main message greatly improve the manuscript. I still think the word "interaction" is somewhat a lost of focus. I suggest the author change that.